# Tumor Lymphocyte Infiltration Is Correlated with a Favorable Tumor Regression Grade after Neoadjuvant Treatment for Esophageal Adenocarcinoma

**DOI:** 10.3390/jpm12040627

**Published:** 2022-04-13

**Authors:** Riad Haddad, Oran Zlotnik, Tal Goshen-Lago, Mattan Levi, Elena Brook, Baruch Brenner, Yulia Kundel, Irit Ben-Aharon, Hanoch Kashtan

**Affiliations:** 1Department of Surgery, Carmel Medical Center, Haifa 3436212, Israel; dr.riad.haddad@gmail.com (R.H.); hkashtan@clalit.org.il (H.K.); 2The Ruth and Bruce Rappaport Faculty of Medicine, Technion, Israel Institute of Technology, Haifa 3525433, Israel; 3Department of Surgery, Rabin Medical Center, Petach Tikva 4941492, Israel; oranzlotnik@gmail.com; 4Division of Oncology, Rambam Health Care Campus, Haifa 3109601, Israel; t_goshenlago@rambam.health.gov.il; 5Sackler Faculty of Medicine, Tel Aviv University, Ramat Aviv, Tel Aviv 6329302, Israel; mattanlevi@gmail.com (M.L.); brennerb@clalit.org.il (B.B.); yuliak@clalit.org.il (Y.K.); 6Department of Pathology, Rabin Medical Center, Petach Tikva 4941492, Israel; elenabr1@clalit.org.il; 7Institute of Oncology, Davidoff Center, Rabin Medical Center, Petach Tikva 4941492, Israel

**Keywords:** esophageal cancer, neoadjuvant treatment, tumor-infiltrating immune lymphocyte, tumor-associated macrophage, tumor regression grade

## Abstract

(1) Background: We aimed to explore the association between neoadjuvant treatment, tumor-infiltrating immune lymphocyte (TIL), and tumor-associated macrophage (TAM) and survival in patients with esophageal adenocarcinoma. (2) Methods: Patients who underwent esophagectomy were divided into three groups according to their treatment modality and tumor regression grade (TRG): (i) surgery-only group (SG), (ii) good responders (GR) group (TRG 0–1), and (iii) bad responders (BR) group (TRG 2–3). We then carried out statistical correlations of the immunofluorescence analysis of the immune infiltrate in the esophageal surgical specimens with several clinical and pathological parameters. In addition, we analyzed The Cancer Genomic Atlas (TCGA) dataset for differences in TILs, TAMs, and protein expression in immune pathways. (3) Results: Forty-three patients (SG—15, GR—13, and BR—13) were evaluated. The highest enrichment of CD3+ (*p* < 0.001), CD8+ (*p* = 0.001) and CD4+ (*p* = 0.009) was observed in the stroma of GR patients. On multivariate analysis, only CD8+ T cell and signet-ring features were independent prognostic factors for overall survival. In TCGA analysis, we identified overexpression of TAM and colony-stimulating factor 1 receptor (CSF-1R). (4) Conclusions: High enrichment of lymphocyte subpopulations in the microenvironment of esophageal adenocarcinoma is associated with a favorable response to neoadjuvant treatment and an improved patient outcome.

## 1. Introduction

Esophageal cancer is the seventh most common malignancy and the sixth leading cause of cancer death globally [1]. It is an aggressive disease that often presents as a locally advanced tumor. Despite advances in diagnostic and therapeutic modalities, the prognosis for esophageal cancer remains dismal. The outcome of the resectable disease has an estimated 5-year survival of 50–55%. Neoadjuvant chemoradiotherapy followed by surgery has become a standard approach in resectable and locally advanced esophageal or gastroesophageal junction cancer [2]. Indeed, perioperative chemotherapy or chemoradiotherapy has shown a significant overall survival benefit of 20–30% at 5 years [2,3]. Previous evidence indicates a correlation between improved survival and the response to neoadjuvant therapy (NAT) [3]. However, recurrence rates following surgery remain high, predominantly in patients with partial or no response to NAT [4]. Moreover, despite evidence indicating a survival benefit following NAT, the degree and rate of the response vary, and to date, there is no effective surrogate biomarker to predict who may benefit from NAT [5,6]. 

Immune checkpoint inhibitors (ICI) have been studied in esophageal cancer in the metastatic setting [7,8] and recently in the adjuvant setting in the pivotal CheckMate 577 study. Findings indicate a survival benefit with the addition of nivolumab, a fully human monoclonal anti-programmed death 1 (PD-1) antibody, compared with surveillance [9].

Furthermore, several components of the tumor microenvironment have been implicated as surrogate biomarkers for the response to ICI in esophageal cancer, including tumor-infiltrating lymphocyte (TIL) and tumor-associated macrophage (TAM) subpopulations [10]. TILs play a key role in inhibiting or supporting the growth and development of tumor cells. Assessment of the tumor immune infiltrate (including subpopulation type, density, distribution within the tumor and peritumoral area) has evolved as a predictive biomarker for tumor progression and response to chemotherapy in several cancer types. 

The immunoscore is a scoring system that summarizes the type, density, function, and localization of immune cells within the tumor, its invasive margin, including the density of T lymphocytes (CD3+) and cytotoxic T lymphocytes (CD8+). Such an immunoscore assay of TILs in colorectal cancer provides an estimate of the risk of recurrence or prolonged survival and is currently proposed to be a component of colorectal cancer staging [11]. A high immunoscore has been found to be significantly associated with a prolonged time to recurrence, disease-free survival, and overall survival in colorectal cancer. Furthermore, chemotherapy has been shown to be significantly associated with survival in patients with a high immunoscore [11]. Lastly, correlations between clinical outcomes and TIL immunoscores have also been demonstrated in other cancers, such as head and neck cancers [12], melanomas [13], and ovarian cancer [14]. In colorectal cancer, using the immunoscore of TILs has become the clinical practice in several settings [11]. 

In esophageal cancer, the prognostic role of TILs remains unclear. Duan et al. [15] found that a high infiltration of CD8+ and Foxp3+ T cells was associated with improved overall survival. However, another study found no prognostic role in a high infiltration of CD3+, CD8+, Foxp3+, and CD45R0+ lymphocytes [16]. A meta-analysis of 30 studies and 5122 patients indicated that despite the heterogeneity of TIL subpopulations in esophageal cancer in different studies, a high infiltrate of generalized TIL and CD8+ and CD4+ T cells may serve as a prognostic marker [17]. 

There is currently an unsolved controversy regarding the correlation between TIL density and complete pathological response after neoadjuvant chemotherapy or chemoradiotherapy in locally advanced esophageal cancer. For example, a study that evaluated TIL density in a cohort of patients who underwent upfront surgery compared with a cohort of patients who received NAT for esophageal cancer found no significant differences in density [18]. Conversely, other studies reported that high levels of TIL density were associated with a significant pathological response to neoadjuvant chemotherapy [19,20]. 

In this study, we aim to characterize immune pathways in esophageal cancer and to further explore the predictive role of immunoscoring for neoadjuvant chemotherapy response and survival in a cohort of patients with operable esophageal cancer. 

## 2. Materials and Methods

### 2.1. Patients

Patients with esophageal adenocarcinoma who underwent surgical resection between 2006 and 2016 were identified from prospectively maintained surgical databases at the Rabin Medical Center. From these, patients for whom formalin-fixed, paraffin-embedded (FFPE) esophageal surgical samples were available for analysis were included in the study. This retrospective study was approved by the Rabin Medical Center Institutional Review Board (IRB) (RMC-0790-16), and the IRB waived patient consent for this study due to its retrospective design. 

Patients were divided into three groups according to their treatment modality and tumor regression grade (TRG) documented in their surgical specimens. All the pathological specimen was reviewed by a pathologist (E.B.), who rated the histopathological response according to tumor regression grade from 1 to 3 as classified by the College of American Pathologist. Patients with an early-stage tumor who underwent surgery without neoadjuvant treatment were defined as the surgery-only group (SG). Patients with a locally advanced tumor and a favorable histopathological response to neoadjuvant treatment (TRG = 0–1) were defined as the good responders group (GR). Patients with a locally advanced tumor and poor response to treatment (TRG = 3) were defined as the bad responders group (BR). Patients with minimal response (residual cancer remaining but with predominant fibroses) (TRG2) were excluded. Patients with clinical data were obtained from the available electronic records and included the patients’ demographics, tumor staging, neoadjuvant chemotherapy, radiation protocol, intraoperative variables, perioperative complications, pathology features, and short and long-term oncological outcomes. Patients were excluded if they underwent definitive chemoradiation or if they had metastatic spread during laparoscopy. 

We kept patient confidentiality throughout the data collection and analysis by replacing protected personally identifiable information with research identification codes (ID codes).

### 2.2. Immunofluorescence

Sections of paraffin-embedded testes were processed as previously described for immunofluorescence [21] with the following primary antibodies: rabbit anti-CD3 (1:100; Cell Marque, Rocklin, CA, USA), goat anti-CD4 (1:100; R&D Systems, Minneapolis, MN, USA), rabbit anti-CD8 (1:100; Cell Marque), goat anti-CD20 (1:100; Abcam, Cambridge, MA, USA), mouse anti-CD45RO (1:100; Cell Marque), mouse anti-CD68 (1:100; Cell Marque), rabbit anti-CD163 (1:100; Cell Marque), goat anti-FoxP3 (1:100; R&D Systems), and mouse anti-pan-cytokeratin (1:100; Cell Marque). We used Hoechst 33280 (1 µg/mL; Sigma Chemicals, St. Louis, MO, USA) for DNA staining, mixed with the following secondary antibodies: Alexa-488 donkey anti-goat (1:200; Abcam), cy3 donkey anti-rabbit (1:200; Jackson Immunoresearch, Baltimore, MD, USA), and Alexa-647 donkey anti-mouse (1:200; Jackson Immunoresearch). The slide was stained with fluorescent markers in different colors that allow a parallel analysis of several markers in the same slide; fluorescence images were photographed using the AxioImager Apotome Microscope fluorescent microscope (CLSM; Carl Zeiss MicroImaging, Oberkochen, Germany) equipped with the Plan-Neofluar 25X objective. Offset calibration of the detector was performed with sections stained with secondary antibodies only. The average number of positively stained cells in each manually selected area were analyzed by Fiji software (National Institutes of Health, Bethesda, Rockville, MD, USA). The data for different areas of the specimen: malignant epithelium, benign epithelium (adjacent to the tumor), malignant stroma, and benign stroma (adjacent to the tumor) were analyzed separately. For each patient, we calculated the mean of positive cells in the benign and malignant stroma and epithelium. 

### 2.3. Characterization of Immune Pathways using The Cancer Genome Atlas (TCGA)

Analyses were performed on the publicly available TCGA dataset. Eligible patients were those defined as having esophageal cancer in the TCGA dataset and who had information on gene expression and reverse-phase protein array (RPPA) analysis available. Analysis of the gene expression pathway was explored using the Gene Set Enrichment Analysis (GSEA) website (software.broadinstitute.org/gsea, accessed on 10 November 2021).

### 2.4. Statistical Analysis 

All statistical analyses were performed using IBM statistics (SPSS) version 25. Continuous variables were summarized with mean ± standard deviation. Categorical variables were presented as numbers and proportions. The correlation between relative densities of lymphocyte subtypes and the included variables was tested using a t-test and analysis of variance (ANOVA) for categorical variables and Pearson correlation for the continuous variables. Overall survival was estimated using Kaplan–Meier curves, and log-rank (Mantel–Cox) tests were used to compare between immune marker expression and disease recurrence or overall survival. Cox proportional hazard regression was used to test the effect of the relative densities of lymphocyte subtypes on overall survival and recurrence-free survival. The method used for multivariate analysis was backward: conditional. Survival was calculated as the time, in months, from the day of surgery until the date of patient death or until the last follow-up date. *p* < 0.05 was considered statistically significant for all tests.

## 3. Results

### 3.1. Clinical Characteristics

Forty-three patients (37 males and 6 females, with a median age of 66 years) were included in the study: 15 (35%) in the SG group, 13 (30%) in the GR group, and 15 (35%) in the BR group. The patients’ characteristics are summarized in Table 1. The gender, age, and tumor location did not differ significantly between the groups. Overall survival was significantly different between the groups (Figure 1).

### 3.2. Pathways Identified in the TCGA Database 

We utilized the data available from the TGCA esophageal tumors to characterize the tumor microenvironment. From these data, we identified pathways related to the tumor immune microenvironment that were correlated with enriched TAM and overexpression of colony-stimulating factor 1 receptor (CSF-1R) (Figure 2), indicating the potential role of these upregulated pathways in esophageal adenocarcinoma.

### 3.3. TILs—All Cohorts

Staining of immune cell surface markers was performed in all 43 esophageal adenocarcinoma specimens, including T lymphocytes (CD3+), B lymphocytes (CD20+), T cytotoxic cells (CD8+), T helper cells (CD4+), T memory cells (CD45RO+), T regulatory cells (FOXP3+), macrophages (CD68+), and M2 macrophages (CD163+) (Figure 3). We first evaluated whether there was a difference in immunostaining between the tumor and the adjacent (benign) stroma. We found no statistically significant differences in the density of immunostaining for CD3+, CD8+, CD45Ro+, Foxp3+, CD20+, and CD163+ cells in the adjacent (benign) stroma and malignant stroma except for CD4+ T cells. The density of CD4+ adjacent stroma was 12.1 ± 8.2% vs. 8.7 ± 8.1% in the malignant stroma (*p* = 0.034). There were no statistically significant differences in the density of immunostaining for CD3+, CD4+, CD8+, CD45R0+, Foxp3+, CD20+, and CD163+ T cells in the adjacent tumor and malignant tumor except for CD8+ T cells (See Appendix A). Due to these findings, we calculated the mean of two density positive staining in the malignant and benign stroma, and from this point on, we referred to this value as “stroma”. We referred to the value of the mean of two density positive staining in the malignant and benign epithelium as “ tumor” (see Appendix A). We classified each of the immune markers as “high expression” when it was above the median threshold value and “low expression” when it was below the median value (see Appendix A). 

The density of positive cell immunostaining of all markers was highest in the GR group, both in the stroma and the tumor (Figure 3 and Figure 4, Appendix A). A significant difference was found in CD3+ (stroma—*p* = 0.003, tumor—*p* < 0.001), CD8+ (tumor—*p* < 0.001), CD4+ (stroma—*p* = 0.013, tumor—*p* = 0.004), and CD45R0+ (tumor—*p* < 0.001) T cells. 

### 3.4. The Correlation between TIL Density and the Response to Neoadjuvant Therapy

Good responders had higher densities of TILs in the stroma and in the tumor in the surgical specimens (Figure 3, Appendix A). The GR group compared to the BR group had significant enrichment in CD3+ (stroma—*p* < 0.001, tumor—*p* < 0.001), CD8+ (stroma—*p* = 0.001, tumor—*p* < 0.001), CD4+ (stroma—*p* = 0.009, tumor—*p* = 0.004), and CD45R0+ (tumor—*p* = 0.014) cells. There was no significant difference in the TIL subpopulations between patients who received neoadjuvant chemoradiotherapy or chemotherapy only. We also documented a positive correlation between lymph node status, pathological staging, and higher densities of TIL subpopulations. Patients without lymph node metastases (N_0_) had significant enrichment of TIL subpopulations compared to patients with lymph node metastasis (N+) in CD3+, CD8+, CD4+, and CD45RO+ cells (Appendix A).

The highest enrichment of CD3+ and CD8+ cells was in Stage I (Appendix A). Patients with lymphovascular invasion (LVI) had an increased density of CD163+ cells in the stroma compared to patients without LVI (25.25 ± 10.0% vs. 14.5 ± 7.4%, respectively; *p* = 0.024). There was no correlation between the density of TIL subpopulations and signet cell features. 

### 3.5. Correlation between TIL Density and Survival

The mean follow-up period was 38.4 ± 26.2 months. A high expression of CD3+, CD8+, CD4+, and CD45R0+ TILs was associated with better disease-free survival and overall survival (Appendix A, Figure 5 and Figure 6).

The univariate analysis found that the immune markers (CD3+, CD8+, CD4, and CD45R0) were associated with better disease-free survival. However, metastatic lymph nodes, Barret features, and signet-ring features were associated with worse disease-free survival (Table 2A). The immune markers (CD3+, CD8+, CD4, and CD45R0) were associated with better overall survival, but metastatic lymph nodes and signet-ring features were associated with worse overall survival (Table 2B). Findings from the multivariate analysis showed that the immune marker CD8+ in the stroma as well as signet features were independent prognostic values for overall survival. 

## 4. Discussion

In this study, we analyzed the density of TIL and TAM subpopulations in patients with esophageal cancer following curative treatment intent. Our results indicate that patients with a favorable pathological response to neoadjuvant chemotherapy had a significantly higher level of TILs (CD3+, CD8+, CD4+, and CD45R0+) in their resected tumors. Moreover, there was a significant association between TIL subpopulation expression and disease-free and overall survival. Of note, in our multivariate analysis, only CD8+ T cells in the stroma were found to be an independent prognostic factor for overall survival. Lastly, the upregulated pathways in the TCGA subset reaffirm the potential role of immune pathways in the pathogenesis and clinical course of esophageal cancer.

Previous studies were inconclusive regarding the density of TILs subpopulation in tumors of patients treated with surgery after neoadjuvant chemotherapy and surgery only. Karstens et al. [22] found a significant increase in CD4 + TIL in peritumoral and tumoral areas from upfront resected EC patients compared to healthy control tissue, but there was no difference in the density of CD4 + TILs between peritumoral and tumoral area. Crumley et al. [18] showed an increased general density of TILs in patients after neoadjuvant chemotherapy compared to surgery alone; however, Noble et al. [19] did not find any significant difference in TILs density between the two groups. Wagener-Ryczek et al. [23] showed an increase in CD3 expression in EC treated by up-front surgery by a factor of three in comparison to normal tissue, but there was a decrease in CD3 expression in patients treated by surgery after neoadjuvant chemotherapy in comparison to upfront surgery but still higher by twofold than normal tissue expression; however, CD8+ T cell was higher three folds in tumor tissue regardless of surgery first or after neoadjuvant chemotherapy. In order to properly assess the effect of chemotherapy on the immune contexture and if the composition of TILs subpopulations can predict the response to chemotherapy in esophageal cancer, it is necessary to compare the density of TILs subpopulation in EUS biopsy at the diagnosis and after neoadjuvant chemotherapy at EUS biopsy or at the resected tumor.

Previous evidence has demonstrated conflicting results for the impact of NAT on TIL density [18,19,22,23]. In this study, we found no difference in the density of TIL subpopulations between the peritumoral and tumoral areas. However, there was a significant increase in the density of TIL subpopulations after neoadjuvant chemotherapy in the GR group compared to the SG group, reflecting a correlation between the density of TILs and the effect of treatment. This correlation was further strengthened by the finding that the lowest TIL density was in the BR group. Additionally, the density of TIL subpopulations (CD3+, CD8+, CD4+, and CD45R0+) in the stroma and the epithelium of the resected tumors was significantly higher in patients who responded to chemotherapy versus non-responders.

Data from previous studies of esophageal cancer showing an association between a high level of TILs and a better clinical and pathological response to neoadjuvant chemotherapy are in concordance with our results. However, those studies referred to a higher general TIL density without specifying TIL subpopulations [18,20]. Nevertheless, Noble et al. [19] showed with a multivariate analysis that higher CD4+ and CD8+ TIL densities are associated with significant tumor response (TRG) after neoadjuvant chemotherapy. This finding was also reported in other cancers, such as breast and ovarian cancer [24], where TIL density at the invasive margin of colorectal liver metastases was shown to have a strong association with chemotherapy efficacy [25]. In rectal cancer, the density of general TILs in resected tumors was shown to have no significant association with the pathological response to neoadjuvant chemoradiotherapy; however, a low density of CD8+ TILs at the invasive margin was significantly associated with a poor pathological response [26].

In the present study, of the TIL subpopulations analyzed, only CD8+ was a significant independent prognostic factor for overall survival. However, it was interesting to see that our data revealed a significant correlation between high levels of CD3+ (stroma, epithelium), CD4+ (stroma, epithelium), CD8+ (stroma, epithelium), and CD45R0 (epithelium) TILs and better disease-free and overall survival. Moreover, the survival analysis revealed that high immunostaining was associated with improved survival and was correlated to the TRG. This supports a previous study that demonstrated that only a minority of patients with high immunostaining had TRG 0–1 [27].

TAMs are an essential component of the tumor microenvironment and play a crucial role in cancer progression. There are two TAM phenotypes: M1 macrophages with an antitumor/proinflammatory function and M2 macrophages (which express the CD163 marker) with a protumor/anti-inflammatory function [28]. CSF-1R act as cell-surface receptors for the cytokines CSF-1 and IL-34 that are secreted by tumor cells and can cause the recruitment of M2-macrophages to support tumorigenesis [29,30,31]. The expression of CSF-1R with the enrichment of TAMs is correlated with a poor prognosis in various types of cancer, including breast, gastric, pancreas, colon, and liver cancer [30]. In the present study, our analysis of the TCGA dataset showed for the first time that there is also a high enrichment of CSF-1R and macrophages in esophageal cancer. We found no correlation between CD68+ and CD163 + TAMs (a marker of M2 macrophages) enrichment and survival. Finally, our TCGA analysis showed a high enrichment of macrophages without the differentiation to M1/M2. However, it is notable that the observed high level of CSF-1R, which is an important receptor for shifting macrophages toward the M2 phenotype, may suggest the importance of M2 macrophages in esophageal cancer tumorigenesis.

Our study has several limitations. The small sample size and the fact that some of the patients had chemotherapy only without radiation are major limitations. Nevertheless, the results demonstrated high statistical significance. Another limitation is the fact that the TIL profile may not represent the original pre-treatment profile, which may have been different. A future prospective study that includes both a pre-treatment evaluation and post treatment may reveal whether this is a major drawback. Due to the fact that diagnostic biopsies usually contain minimal material, immunostaining is unfeasible. Our study’s findings indicate a correlation between specific TILs and the patients’ response to NAT as well as their survival. Due to the evolving role of immunotherapy in the adjuvant setting, it is prudent to explore whether the immune landscape that correlates to chemotherapy and chemoradiation may also predict response to immunotherapy for better patient selection.

## 5. Conclusions

Our results demonstrate that a high enrichment of lymphocyte subpopulations in the microenvironment of esophageal adenocarcinoma tumors is correlated with both a favorable tumor regression after neoadjuvant treatment and with improved survival. This proposed signature may potentially serve for tailored treatment and better patient selection for NAT. This would be especially true in early disease cases where there are several therapeutic options, including perioperative chemotherapy, neoadjuvant chemotherapy, and upfront surgery. Further prospective studies are warranted to determine how immune-based classification could guide clinical decision making in esophageal cancer.

## Figures and Tables

**Figure 1 jpm-12-00627-f001:**
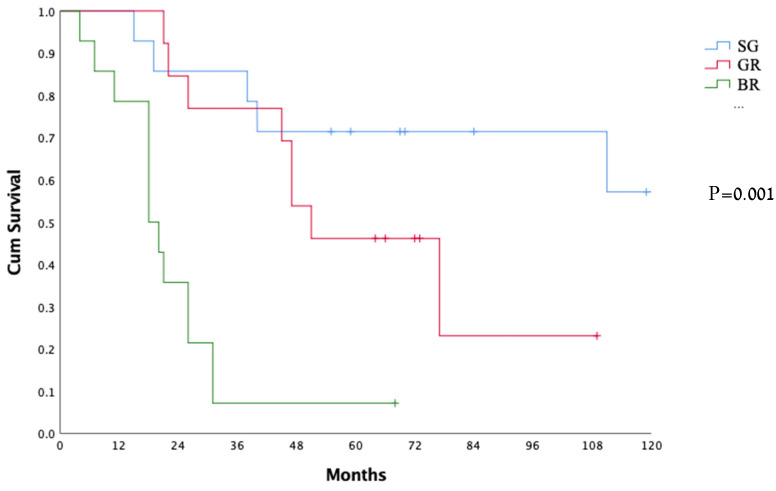
Kaplan–Meier curves of overall survival of all the cohort. SG—surgery group; GR—good responders; BR—bad responders.

**Figure 2 jpm-12-00627-f002:**
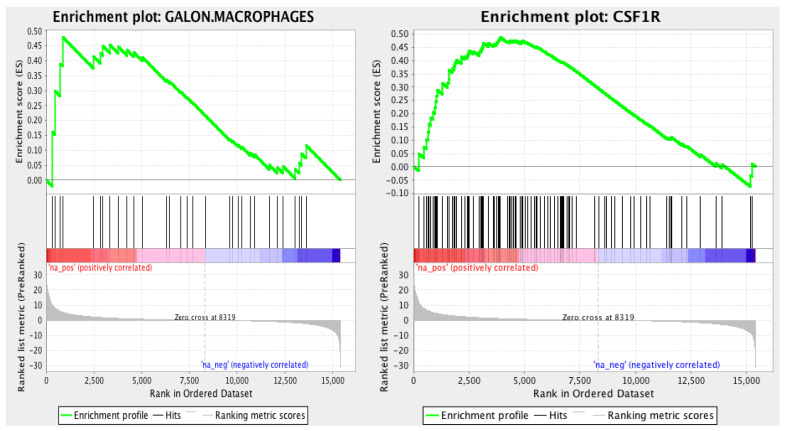
TGCA gene expression pathway analysis, high enrichment of TAM, and overexpression of colony-stimulating factor 1 receptor (CSF-1R).

**Figure 3 jpm-12-00627-f003:**
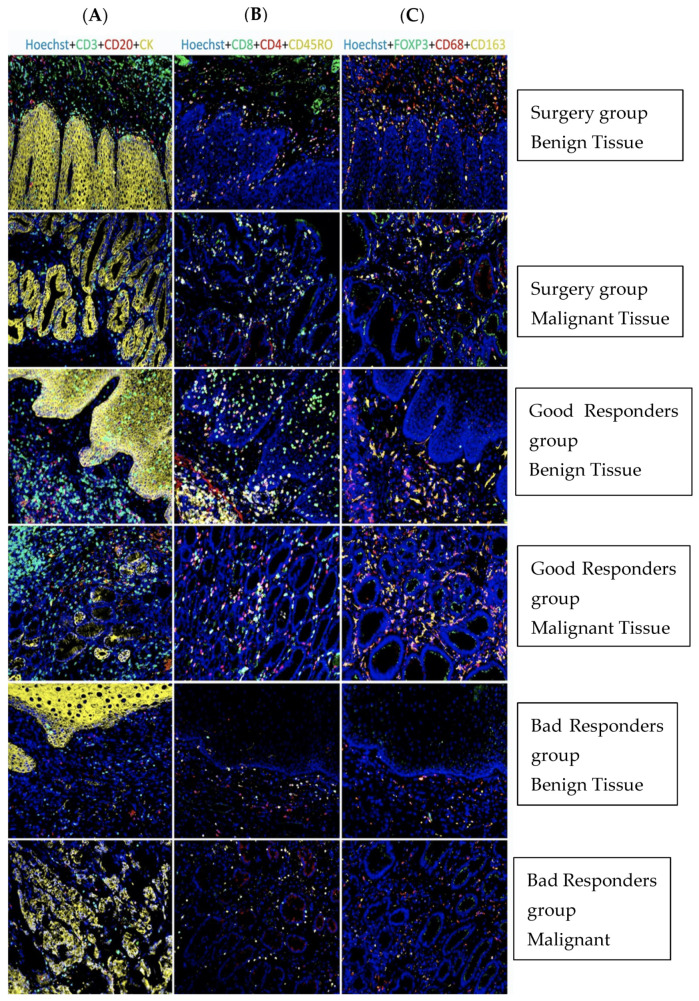
Representative graphic quantification of tumor-infiltrating lymphocytes in the tumor microenvironment of different groups in the cohort. Column (**A**): CD3 + TIL is marked in green, CD20 + TIL in red, and CK-cytokeratin is marked in yellow; Column (**B**): CD8 + TIL is marked in green, CD4 + TIL in red, and CD45RO + TIL in yellow; Column (**C**): Foxp3 + TIL is marked in green, CD68+ in red, and CD163+ in yellow.

**Figure 4 jpm-12-00627-f004:**
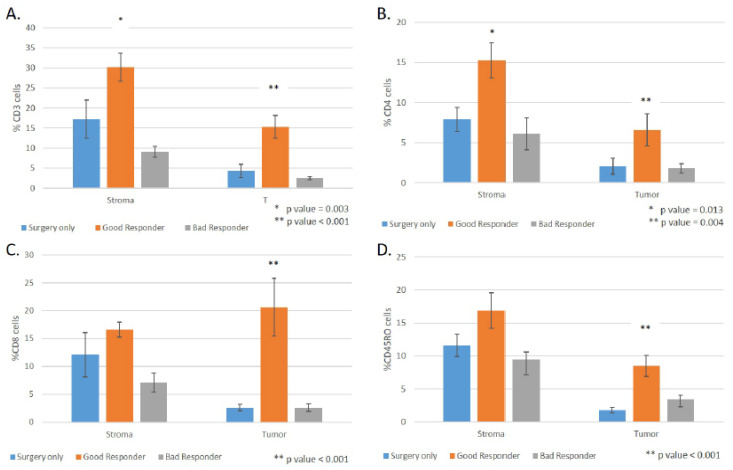
Densities of different lymphocytes subpopulations quantified from immunohistochemistry slides and compared between the different groups in the cohort: surgery only (blue), good responders (orange), and bad responders (grey). The lymphocytes subpopulations presented are: (**A**)**.** T cells (CD3) (**B**). T helper cells (CD4), T cytotoxic cells (CD8) (**C**)**.** T cytotoxic cells (CD8) (**D**)**.** T memory cells (CD45RO). Data are presented as mean ± standard deviation; *p*-value was assessed by ANOVA test.

**Figure 5 jpm-12-00627-f005:**
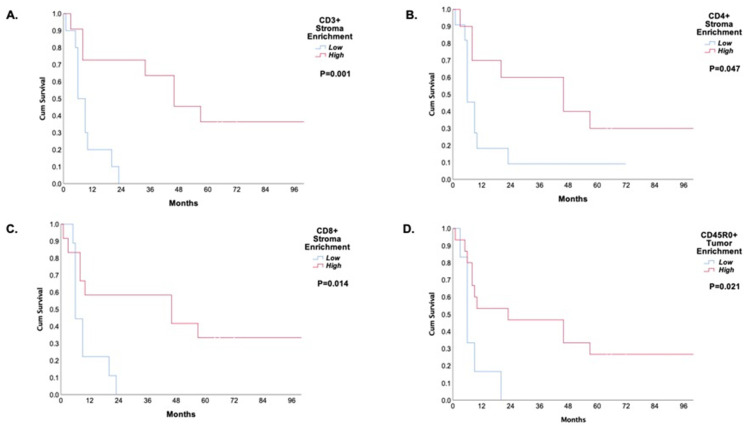
Kaplan–Meier curves of five-year disease-free survival of the neoadjuvant treated patients based on the tumor-infiltrating immune lymphocyte subpopulations. (**A**) CD3+ TIL stroma; (**B**) CD4 + TIL stroma; (**C**) CD8+ TIL stroma; (**D**) CD45RO+ TIL tumor.

**Figure 6 jpm-12-00627-f006:**
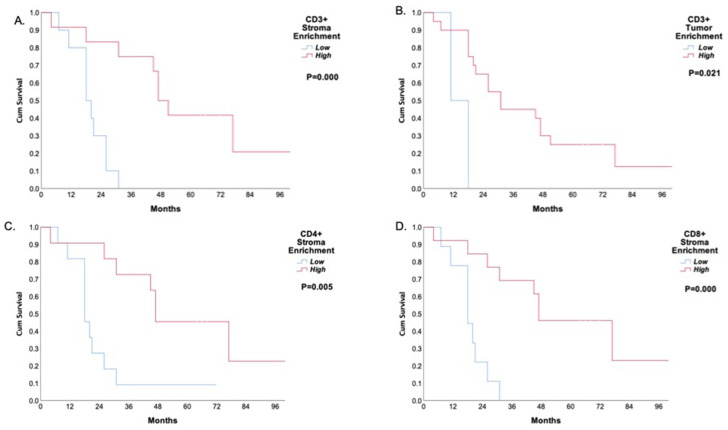
Kaplan–Meier curves of five-year overall survival of the neoadjuvant treated patients based on the tumor-infiltrating immune lymphocyte subpopulations. (**A**) CD3+ TIL stroma; (**B**) CD3+ TIL tumor; (**C**) CD4 + TIL stroma; (**D**) CD8+ TIL stroma; (**E**) CD8+ TIL tumor.

**Table 1 jpm-12-00627-t001:** Patients’ clinical characteristics.

KERRYPNX	Total *n* = 43	Surgery Only *n* = 15 (35%)	Good Responders *n* = 13 (30%)	Bad Responders *n* = 15 (35%)	*p*-Value
Gender					0.21
Male	36 (88%)	11 (73%)	12 (92%)	14 (93%)	
Female	5 (12%)	4 (27%)	1 (8%)	1 (7%)	
Age (years)	66 ± 11	70 ± 7	67 ± 11	61 ± 12	0.09
Tumor location					0.62
GEJ	27 (63%)	10 (67%)	7 (54%)	10 (67%)	
Lower	15 (35%)	5 (33%)	5 (38%)	5 (33%)	
Mid	1 (2%)	0 (0%)	1 (8%)	0 (0%)	
Clinical stage I II III Missing data	13 (30%) 9 (21%) 18 (42%) 3 (7%)	12 (93%) 1 (7%)	1(7%) 4 (27%) 8 (53%) 2 (13%)	0 (0%) 4 (27%) 10 (71%) 1 (7%)	0.001
Neoadjuvant treatment Chemoradiation Chemotherapy	17 (60%) 11 (40%)		11 (85%) 2 (15%)	6 (40%) 9 (60%)	0.02
Erbitux Yes No	4 (9%) 39 (91%)		1 (8%) 12 (92%)	3 (20%) 12 (80%)	0.14
T					0.001
CR	5 (12%)		5 (39%)	0 (0%)	
1	12 (30%)	10 (67%)	3 (23%)	0 (0%)	
2	6 (14%)	3 (20%)	2 (15%)	1 (7%)	
3	19 (44%)	2 (13%)	3 (23%)	14 (93%)	
LN extracted	11 ± 6	11 ± 4	8 ± 5	15 ± 7	0.004
N					0.001
N0	25 (58%)	12 (80%)	12 (92%)	1 (7%)	
N+	18 (42%)	3 (20%)	1 (8%)	14 (93%)	
Stage					0.001
I	22 (51%)	12(80%)	10 (77%)	0 (0%)	
II	3 (7%)	0 (0%)	2 (15%)	1 (7%)	
III	18 (42%)	3 (20%)	1 (8%)	14 (93%)	
Signet features					0.2
Yes	4 (9%)	0 (0%)	1 (8%)	3 (20%)	
No	34 (79%)	11 (73%)	12 (92%)	11 (67%)	
Missing data	5 (12%)	4 (27%)	0 (0%)	2 (13%)	
TRG 0 1 3	5 (18%) 8 (29%) 15 (53%)		5 (39%) 8 (61%)	15 (100%)	0.001
Vascular invasion					0.07
Yes	11(26%)	1 (7%)	4 (31%)	6 (40%)	
No	30 (70%)	14 (93%)	8 (62%)	8 (53%)	
Missing data	2 (4%)		1 (7%)	1 (7%)	
Neural invasion Yes No Missing data	9 (21%) 33 (77%) 1 (7%)	3 (20%) 12 (80%)	2 (15%) 10 (77%) 1 (8%)	4 (27%) 11 (73%)	0.9
Survival (months)	52 ± 37	77 ± 43	55 ± 25	22 ± 15	0.001

GEJ—esophagogastric junction; TRG—tumor regression grade; CR—complete response; LN—lymph node; N—node.

**Table 2 jpm-12-00627-t002:** Cox regression analysis for disease-free survival (**A**) and overall survival (**B**). LN—lymph node; mets—metastases; T—T stage; CR—complete response.

	Univariate Analysis	Multivariate Analysis
Characteristics	HR (95%CI)	*p*-Value	HR (95%CI)	*p*-Value
CD3 stroma (low vs. high)	6.74 (1.79–25.4)	0.005		
CD8 stroma (low vs. high)	3.59 (1.16–11.13)	0.026		
CD8 tumor (low vs. high)	3.1 (0.99–9.56)	0.05		
CD4 stroma (low vs. high)	2.88(1.0–8.30)	0.049		
CD45R0 tumor (low vs. high)	3.2 (1.05–9.78	0.04		
Barret features (no vs. yes)	0.19 (0.05–0.80)	0.023		
Signet features (no vs. yes)	0.23 (0.07–0.79)	0.02		
LN (negative vs. positive mets)	0.3 (0.11–0.84)	0.022		
T (CR vs. T1 − T3)	0.4 (0.11–1.4)	0.15		
CD3 stroma (low vs. high)	8.43 (2.13–33.3)	0.002		
CD3 tumor (low vs. high)	5.36 (1.02–28.1)	0.047		
CD8 stroma (low vs. high)	6.93 (1.99–24.15)	0.002	27.3 (3.2–233.8)	0.002
CD8 tumor (low vs. high)	6.12 (1.67–22.5)	0.006		
CD4 stroma (low vs. high)	4.1 (1.4–12.11)	0.011		
CD45R0 tumor (low vs. high)	4.9 (1.43–16.17)	0.011		
Signet features (no vs. yes)	0.27 (0.08–0.89)	0.031	0.027 (0.002–0.3)	0.003
LN (negative vs. positive mets)	0.19 (0.07–0.50)	0.001		
T (CR vs. T1 − T3)	0.40 (0.12–1.4)	0.15		
Chemoradiation vs. chemotherapy	1.84 (0.76–4.44)	0.18

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
