# Peer review of "Tumor Lymphocyte Infiltration Is Correlated with a Favorable Tumor Regression Grade after Neoadjuvant Treatment for Esophageal Adenocarcinoma"

_jpm, 2022, doi:10.3390/jpm12040627_

Round 1
Reviewer 1 Report
This is a retrospective review of 43 patients with localized esophageal cancer, who underwent surgery alone (15) vs. pre-operative chemotherapy (11) or chemoradiation (17). Immunofluorescence (IF) was used to analyze the infiltration of immune cells into the tumor and surrounding stroma. The results suggest that patients with good vs. bad response had a higher density of CD3, CD8, CD4 and CD45R0 cells in the tumor and stroma. In univariate analysis, these markers were also associated with improved DFS and OS. In multivariate analysis, only CD8 cells in the stroma were correlated with improved OS.
Ultimately, these results are hypothesis-generating and argue that such analyses merit ongoing evaluation.
My comments:
- I don’t see any clear value to the case report of the patient who developed GBS after chemoRT. The hypothesis that this represents an immune reaction to chemoRT and that this immune reaction contributed to the pathologic complete response is unsubstantiated. This patient underwent surgery at least 4 months after the completion of chemoRT, which is a confounding variable in the immunofluorescence analysis. I would recommend removing this patient entirely from the manuscript, unless the authors are able to provide further justification to include it as a noteworthy anecdote.
- I suspect that part of the reason for the mixed results of similar analyses is the variable methodology used to perform the IF or IHC staining of immune cells. How was this methodology developed and validated in this study? For patients with a pathologic complete response, how was “tumor” tissue identified and selected?
- The authors have appropriately highlighted the limitations of this study. In their next analysis, I would recommend restricting their analysis to patients who receive either chemoRT or chemo alone so as to better standardize the patient population. As much as possible and understanding the challenges involved, baseline and surgical specimens should be analyzed since this is a more appropriate control than patients who underwent surgery alone.
- Can the authors explain how a signature generated in surgical specimens can be used to select patients for neoadjuvant therapy?
Author Response
We want to thank the editorial board and the reviewers for their careful review of our submitted manuscript. Please find below a point-by-point response addressing their valuable comments. Changes in the manuscript were made as needed and tracked.
My comments:
- I don’t see any clear value to the case report of the patient who developed GBS after chemoRT. The hypothesis that this represents an immune reaction to chemoRT and that this immune reaction contributed to the pathologic complete response is unsubstantiated. This patient underwent surgery at least 4 months after the completion of chemoRT, which is a confounding variable in the immunofluorescence analysis. I would recommend removing this patient entirely from the manuscript, unless the authors are able to provide further justification to include it as a noteworthy anecdote.
Response : after reconsideration we agree with the reviewer and we remove the case report through the manuscript (including: materials and methods, results, discussion, references and supplementary figure 1 ).
- I suspect that part of the reason for the mixed results of similar analyses is the variable methodology used to perform the IF or IHC staining of immune cells. How was this methodology developed and validated in this study? For patients with a pathologic complete response, how was “tumor” tissue identified and selected
Response: we perform Immunofluorescence analysis. Sections of paraffin-embedded slide was stained with fluorescent markers in different colors that allow parallel analysis of several immune markers in the same slide (line 135-136).
We add “All the pathological specimen was reviewed by a pathologist (E.B.), who rated the histopathological response according to tumor regression grade 1 to 3 as classified by the College of American Pathologist” (line 106-109).
- The authors have appropriately highlighted the limitations of this study. In their next analysis, I would recommend restricting their analysis to patients who receive either chemoRT or chemo alone so as to better standardize the patient population. As much as possible and understanding the challenges involved, baseline and surgical specimens should be analyzed since this is a more appropriate control than patients who underwent surgery alone.
Response: this one of the main limitations of the study .“Another limitation is the fact that the TIL profile may not represent the original pre-treatment profile, which may have been different. A future prospective study that includes both a pre-treatment evaluation and post treatment may reveal whether this is a major drawback” (line 496-499)
- Can the authors explain how a signature generated in surgical specimens can be used to select patients for neoadjuvant therapy?
Response : Only a future prospective study that includes both a pre-treatment evaluation (EUS biopsy ) and post treatment specimen may reveal the effect of the TIL and TAM on the response to neoadjuvant therapy.
Reviewer 2 Report
In this manuscript Haddad R. and colleagues describe the association between tumor lymphocyte infiltration and favourable outcome of oesophageal cancer patients after neoadjuvant chemotherapy.
The study is of interest, but it is predominately limited by the lack of novelty. Several previous reports have addressed similar questions with analogous conclusions (PMID: 31981293, 31366557).
Author Response
We want to thank the editorial board and the reviewers for their careful review of our submitted manuscript. Please find below a point-by-point response addressing their valuable comments. Changes in the manuscript were made as needed and tracked.
In this manuscript Haddad R. and colleagues describe the association between tumor lymphocyte infiltration and favourable outcome of oesophageal cancer patients after neoadjuvant chemotherapy.
The study is of interest, but it is predominately limited by the lack of novelty. Several previous reports have addressed similar questions with analogous conclusions (PMID: 31981293, 31366557).
Response: We agree with the reviewer that several reports address the effect of Tumor‐infiltrating immune lymphocyte and tumor-associated macrophage on esophageal cancer, however these reports including even 5122 patients meta-analyzes have several limitations. First, the main problem is the heterogeneity in patient selection, most of these reports include only patients with esophageal squamous cell carcinoma (including the two referred reports ) , or combined with adenocarcinoma. Another limitation is the differences in the TIL subpopulation analysis that was performed including a number of markers, the cut-off points for high vs low enrichment of each marker.
In our study design, we aim to characterize the predictive role of immune markers for neoadjuvant chemotherapy response depending on the TRG in the surgical specimen .
Reviewer 3 Report
Gaddd and co-workers describe the relationship between lymphocyte infiltration and tesophageal adenomcarcinoma tumor regression after treatment.
Major comments
Introduction:
Please clarify what this study adds to the 5122 patient meta-analysis previously described (line 8) and add this to the aim (line 93-95)
The methods are not adequetely described , proteome analysis, rna sequencing, genome analysis (line 282-286) is missing, It is not described how the TCGA dataset was analysed (or what was analysed) GSEA analysis is not described (e.g. compared to what erference, whatwere the settings ,waht was explored (which genes) etc
the results of genome, transcriptome and genomic analysis are not described. has this been done for all patients or for this unique cpatient specificallly? How do the survival plots relate to the genetic make-up of the tummours (e.g. are driver mutations , number of genomic abberations etc equally dived over the groups?)
line 199 and further: pathways are identified (how?) and where do the two genes come from (TAM and CSF1R )
line 207-220: hardely any statistical differences between comparisons, line 220-228: almost all groups are statistical different. Pleasy clarify why this type of analysis is approriate. Why were there no differences in the first analysis (outliers) and do you exclude these (probably) in the binning step. What is the releveance of outliers to the interpretation of the results.
Discussion
There is little mention at all about the impact of different driver mutations, different classes of tumours etc on lymphocyte infiltrate and subsequent survival.
Minor comments
abstract
please specifiy the groups in more detail (as in the methods)
results
there is a large difference between males and females included in the study, doe this reflect the patient population
line 44-45 2x dismal, please reformulate one of them.
Figure 3: i find it hard to see differences between the groups and relate it to figure 4. How many fields are counted (or is the entire slide counted)? How is the distribution of the infiltrate over the tumour/stroma etc
Author Response
We want to thank the editorial board and the reviewers for their careful review of our submitted manuscript. Please find below a point-by-point response addressing their valuable comments. Changes in the manuscript were made as needed and tracked.
Gaddd and co-workers describe the relationship between lymphocyte infiltration and tesophageal adenomcarcinoma tumor regression after treatment.
Major comments
Introduction:
Please clarify what this study adds to the 5122 patient meta-analysis previously described (line 8) and add this to the aim (line 93-95)
Response: We agree with the reviewer that several reports address the effect of Tumor‐infiltrating immune lymphocyte and tumor-associated macrophage on esophageal cancer, however these reports including even 5122 patients meta-analyzes have several limitations. First, the main problem is the heterogeneity in patient selection, most of these reports include only patients with esophageal squamous cell carcinoma , or combined with adenocarcinoma. Another limitation is the differences in the TIL subpopulation analysis that was performed including number of markers, the cut-off points for high vs low enrichment of each marker.
In our study design we aim to to characterize the predictive role of immune markers for neoadjuvant chemotherapy response depend on the TRG in the surgical specimen .
The methods are not adequetely described , proteome analysis, rna sequencing, genome analysis (line 282-286) is missing, It is not described how the TCGA dataset was analysed (or what was analysed) GSEA analysis is not described (e.g. compared to what erference, whatwere the settings ,waht was explored (which genes) etc. the results of genome, transcriptome and genomic analysis are not described.
has this been done for all patients or for this unique cpatient specificallly? How do the survival plots relate to the genetic make-up of the tummours (e.g. are driver mutations , number of genomic abberations etc equally dived over the groups?)
line 199 and further: pathways are identified (how?) and where do the two genes come from (TAM and CSF1R )
Response : after reconsideration we agree with the reviewer 1 and we remove the case report of the unique patient. We did perform only for this patient a full genomic and proteomic analysis of his tumor`s microenvironment and RNA sequencing.
For all other 43 patients with esophageal adenocarcinoma we perform only immunofluorescence analysis for TIL and TAM subpopulation. We did not perform any genomic analysis for this cohort of patients.
Characterization of immune pathways using The Cancer Genome Atlas (TCGA), Analyses were performed on the publicly available TCGA dataset, we look specifically for immune pathways in esophageal cancer. From this data, we identified enrichment of tumor-associated macrophage (TAM) and over expression of colony-stimulating factor 1 receptor (CSF-1R).
line 207-220: hardely any statistical differences between comparisons, line 220-228: almost all groups are statistical different. Pleasy clarify why this type of analysis is approriate. Why were there no differences in the first analysis (outliers) and do you exclude these (probably) in the binning step. What is the releveance of outliers to the interpretation of the results.
Response: The data for immunostaining was analyzed separately from “ different areas of the specimen: malignant epithelium, benign epithelium (adjacent to the tumor), malignant stroma and benign stroma (adjacent to the tumor)”. (line 145-147).
We firstly evaluated whether there was a difference in immunostaining between the tumor and the adjacent (benign) stroma. (line 232-234). We found no significant difference in the density of immune staining in the malignant and benign stroma. (234-236).
Due to these findings, we calculated the mean of two density positive staining in the malignant and benign stroma and from this point on we referred to this value as “stroma”. We referred to the value of the mean of two density positive staining in the malignant and benign epithelium as “ tumor”. (line 241-244).
Discussion
There is little mention at all about the impact of different driver mutations, different classes of tumours etc on lymphocyte infiltrate and subsequent survival.
Response : we remove the case report of the unique patient. We did perform only for this patient a full genomic and proteomic analysis of his tumor`s microenvironment and RNA sequencing.
For all other 43 patients with esophageal adenocarcinoma we perform only immunofluorescence analysis for TIL and TAM subpopulation. We did not perform any genomic analysis for this cohort of patients.
Minor comments
abstract
please specifiy the groups in more detail (as in the methods)
Response : Due to the limitation on the number of words in the abstract we could not elaborate further.
results
there is a large difference between males and females included in the study, doe this reflect the patient population
Response: this was our patientpopulation
line 44-45 2x dismal, please reformulate one of them.
We delete dismal on line 45
Figure 3: i find it hard to see differences between the groups and relate it to figure 4. How many fields are counted (or is the entire slide counted)? How is the distribution of the infiltrate over the tumour/stroma etc
Response: The data for immunostaining was analyzed separately from “ different areas of the specimen: malignant epithelium, benign epithelium (adjacent to the tumor), malignant stroma and benign stroma (adjacent to the tumor)”. (line 145-147).
We firstly evaluated whether there was a difference in immunostaining between the tumor and the adjacent (benign) stroma. (line 232-234). We found no significant difference in the density of immune staining in the malignant and benign stroma. (234-236).
Due to these findings, we calculated the mean of two density positive staining in the malignant and benign stroma and from this point on we referred to this value as “stroma”. We referred to the value of the mean of two density positive staining in the malignant and benign epithelium as “ tumor”. (line 241-244).
Reviewer 4 Report
In the manuscript, the authors described a correlation between tumor lymphocytes infiltration (TILs), tumor associated macrophages (TAMs) and prognosis after neoadjuvant treatment in easophageal adenocarcinoma. Significantly higher level of TILs is related to a good prognosis. This can be developed to be a method for neoadjuvant treatment evaluation.
In the introduction part, the authors aims to characterize immune pathways in esophageal cancer. TCGA analysis was performed and the authors reported that the enrichment of TAM and overexpression of colony-stimulating factor 1 receptor (CSF-1R) is correlated with a poor prognosis in easophageal adenocarcinoma. Unfortunately, no further study on these two factors was done in this manuscript. TILs were put as a center of this study, not TAM and CSF1-R.
This manuscript suffers from discontinuation of the study. Below are my comments
Major points:
- More investigation on immune pathways analysis should be performed. Staining of CSF1-R and further analysis of macrophage phenotype in patient's samples should be examined. This will confirm the finding from TCGA experiment.
- In figure 4. only T cells expressions were quantified from immunohistochemistry and shown, but not macrophage.
- In discussion part (Line 402-407), the authors mentioned that a high enrichment of CD68+ TAMs was 26% compared to 54% for patients with a low CD68+ TAMs enrichment. Which figure or table did the author take this data from?
Minor points:
- Figure 1, type of statistic analysis should be described in the legend.
- Legend of Figure 3, word "Colum" should be "column"
- Legend of Figure 3, Column A, definition of "CK" is missing.
- Figure 4, type of statistic analysis should be described in the legend.
Author Response
We want to thank the editorial board and the reviewers for their careful review of our submitted manuscript. Please find below a point-by-point response addressing their valuable comments. Changes in the manuscript were made as needed and tracked.
In the manuscript, the authors described a correlation between tumor lymphocytes infiltration (TILs), tumor associated macrophages (TAMs) and prognosis after neoadjuvant treatment in easophageal adenocarcinoma. Significantly higher level of TILs is related to a good prognosis. This can be developed to be a method for neoadjuvant treatment evaluation.
In the introduction part, the authors aims to characterize immune pathways in esophageal cancer. TCGA analysis was performed and the authors reported that the enrichment of TAM and overexpression of colony-stimulating factor 1 receptor (CSF-1R) is correlated with a poor prognosis in easophageal adenocarcinoma. Unfortunately, no further study on these two factors was done in this manuscript. TILs were put as a center of this study, not TAM and CSF1-R.
This manuscript suffers from discontinuation of the study. Below are my comments
Major points:
- More investigation on immune pathways analysis should be performed. Staining of CSF1-R and further analysis of macrophage phenotype in patient's samples should be examined. This will confirm the finding from TCGA experiment.
Response : we did perform immunostaining analysis (CD 68+ and CD163+) for Tumor associated macrophage (TAM) (line 232, Supplementary Table 1).
- In figure 4. only T cells expressions were quantified from immunohistochemistry and shown, but not macrophage.
Response : In Figure 4, we showed only the immune markers that are with a significant statistical difference between the groups. The macrophages markers (CD68+ and CD163+) were not significant statistical different between the groups (data are shown in supplementary table B).
- In discussion part (Line 402-407), the authors mentioned that a high enrichment of CD68+ TAMs was 26% compared to 54% for patients with a low CD68+ TAMs enrichment. Which figure or table did the author take this data from?
Response : we delete “In our patient cohort we observed the negative effect of enriched CD68+ TAMs (general macrophages including M1 macropaghes and subtype M2b macrophages) on survival; the 5-year survival for patients with a high enrichment of CD68+ TAMs was 26% compared to 54% for patients with a low CD68+TAMs enrichment (P=0.034).”
It was our mistake this data belong to patients with esophageal squamous cell cancer.
Minor points:
- Figure 1, type of statistic analysis should be described in the legend.
Response : we add “Kaplen-Meier curves of” (line 211)
- Legend of Figure 3, word "Colum" should be "column"
Response : we replace “colum” by “column” ( line 273-276)
- Legend of Figure 3, Column A, definition of "CK" is missing.
Resopnse: we add “CK-cytokertin is marked in yellow” (line 275)
- Figure 4, type of statistic analysis should be described in the legend.
Response : we add “Data are presented as mean+standard deviation, P value was assested by ANOVA test.
Round 2
Reviewer 2 Report
No additional comments.
Author Response
Attached below
Reviewer 3 Report
Authors have addressed some of the concerns and suggestions raised in their rebuttal letter. Authors have decided to remove the unique case instead of describing in the methods how these analysis were done.
Several questions remain, how were the gene set enrichment analysis performed, how was the pathway enrichment analysis performed, rephrasing of the hypothesis in the text to better match the answer given in the rebuttal letter, does their cohort sex distribution match that of earlier published studies (yes this is your cohort, but there is a skewed sex distribution, is it always like this) etc etc.
Author Response
Attached below.
Reviewer 4 Report
The authors satisfactory addressed all the questions.
Author Response
Attached below.